# Longitudinal Changes in BDNF and MMP-9 Protein Plasma Levels in Children after Cochlear Implantation

**DOI:** 10.3390/ijms24043714

**Published:** 2023-02-13

**Authors:** Monika Matusiak, Dominika Oziębło, Monika Ołdak, Emilia Rejmak, Leszek Kaczmarek, Henryk Skarżyński

**Affiliations:** 1Oto-Rhino-Laryngosurgery Clinic, Institute of Physiology and Pathology of Hearing, M Mochnackiego 10, 02-042 Warsaw, Poland; 2World Hearing Centre, Mokra 17, 05-830 Nadarzyn, Poland; 3Department of Genetics, Institute of Physiology and Pathology of Hearing, M Mochnackiego 10, 02-042 Warsaw, Poland; 4BRAINCITY, Nencki Institute of Experimental Biology, L Pasteura 3, 02-093 Warsaw, Poland

**Keywords:** congenital deafness, cochlear implantation, neuroplasticity, MMP-9, BDNF, auditory development

## Abstract

Congenitally deaf children who undergo cochlear implantation before 1 year of age develop their auditory skills faster than children who are implanted later. In this longitudinal study, a cohort of 59 implanted children were divided into two subgroups according to their ages at implantation—below or above 1 year old—and the plasma levels of matrix metalloproteinase-9 (MMP-9), brain-derived neurotrophic factor (BDNF), and pro-BDNF were measured at 0, 8, and 18 months after cochlear implant activation, while auditory development was simultaneously evaluated using the LittlEARs Questionnaire (LEAQ). A control group consisted of 49 age-matched healthy children. We identified statistically higher BDNF levels at 0 months and at the 18-month follow-ups in the younger subgroup compared to the older one and lower LEAQ scores at 0 months in the younger subgroup. Between the subgroups, there were significant differences in the changes in BDNF levels from 0 to 8 months and in LEAQ scores from 0 to 18 months. The MMP-9 levels significantly decreased from 0 to 18 months and from 0 to 8 months in both subgroups and from 8 to 18 months only in the older one. For all measured protein concentrations, significant differences were identified between the older study subgroup and the age-matched control group.

## 1. Introduction

### 1.1. Variability in Auditory Development after a Cochlear Implantation

Auditory development after congenital deafness treatment with cochlear implantation is a dynamic, multifaceted process, and linguistic outcomes after the procedure vary widely. There are certain recognized factors, such as the age at which the procedure is performed, that are known to be major determinants [1,2,3,4]. This is understandable given the large body of studies showing that the absence of auditory input to the juvenile brain has far-reaching consequences, largely because of a gradual and irreversible loss of neuronal plasticity from birth onwards [2]. However, even when cochlear implantation is performed within the limited time window of high plasticity—that is, within the first 2–3 years of life—the functional results of cochlear implantation are very diverse [1,2,3,4,5,6]. That means it is difficult to identify—before an implant is installed—the odds of that child going on to have a poor outcome. A biomarker that could help assess the chances of a good cochlear implantation outcome would be very helpful. We already know that the trajectories of auditory development in children implanted early in life differ markedly from the trajectories of those implanted later on, indicating that there is a faster increase in auditory skills when a cochlear implant (CI) is activated very early compared to when it is activated after 1 year of life [3,4,5,7,8,9,10].

### 1.2. Molecular Basis of Neuroplasticity after Sensory Deprivation

Congenital deafness means that the auditory system is essentially unstimulated from birth onwards. This has far-reaching consequences, affecting the neurophysiology of the brain at multiple levels, such as neuronal connectivity, the ability to form synapses, and cortical maturation [2,6,11]. These processes have molecular underpinnings in a cascade of protein mechanisms that regulate the strength of neuronal connections [2,6,11]. It is known that one of the molecules involved in neuronal connectivity after deafness treatment is matrix metalloproteinase-9 (MMP-9) [12]. MMP-9 plays a critical role in synaptic transmission: in neuronal cell bodies it has been localized predominantly to dendritic spines, which harbor the main mass of excitatory synapses [13,14,15,16,17,18,19,20]. One possible target for MMP-9 could be another protein with a well-documented and pivotal role in neuronal plasticity—brain-derived neurotrophic factor (BDNF) [21,22,23,24]. This molecule is secreted in the form of pro-BDNF, which is cleaved to produce mature BDNF, and MMP-9 has been implicated in this conversion [25] The pro-BDNF/BDNF ratio has been analyzed in clinical samples of, e.g., schizophrenia [23,26].

Activated BDNF is a key factor in the molecular cascade of long-term potentiation (LTP), a process involved in cognition, memory, and learning [21,26,27,28,29]. A body of research has shown that, despite very complex, tight regulation of MMP-9 and BDNF expression, abnormal blood levels of both these proteins correspond with the development of many clinical conditions [14,19,30]. These findings have been documented in stroke (and its recovery) [19], depression and how well it responds to treatment [31], mood disorders [32], the duration of epilepsy [28,29,33], and success in treating deafness with cochlear implantation [34]. In our first paper reporting the results of a prospective cohort study of a large group (N = 61) of congenitally deaf CI children, we documented the roles of MMP-9 and BDNF in auditory development after cochlear implantation. This work revealed that the MMP-9 plasma level measured at implantation predicts auditory development 18 months later [12]. The same strong prediction emerged in a subgroup of children implanted in the second year of life, whereas in a group implanted before their first birthdays, MMP-9 measured at implantation provided no predictive value [12]. This difference in the predictive value of MMP-9 suggests that neuronal rearrangement after the delivery of electrical pulses to the auditory system may involve different molecular machinery in older children than in younger ones [35]. This has motivated us to analyze the profiles of auditory development of these two subgroups over 18 months while simultaneously measuring the plasma profiles of MMP-9, BDNF, and pro-BDNF protein levels and the pro-BDNF/BDNF ratio.

To the best of our knowledge no other group has published data identifying the molecular driver for auditory development that is set in motion by the electrical stimuli delivered by a CI.

### 1.3. Aim of the Study

The aims of the study were two-fold: (i) to test the assumption that the plasma levels of MMP-9, BDNF, and pro-BDNF (as well as the pro-BDNF/BDNF ratio) measured at three time points in a group of implanted children whose CIs were activated before 1 year old were significantly different from the same measures made at the same time points in a second group of implanted children whose CIs were activated after 1 year old and (ii) to elucidate whether the same parameters measured at the end-point (after 18 months of CI use) were similar to those in an age-matched control group of normal-hearing children.

## 2. Results

### 2.1. LEAQ Scores; MMP-9, BDNF, and Pro-BDNF Plasma Levels; and Pro-BDNF/BDNF Ratio in the Study Group

In the whole study group, the mean LEAQ score collected at cochlear implantation was 6.37 (min. 0, max. 28, SD 7.17), the mean plasma level of MMP-9 measured at cochlear implantation was 233.08 ng/mL (min = 31.14, max = 769.67, SD = 129.03), the mean plasma level of BDNF was 2.30 ng/mL (min = 0.25, max = 12.18, SD = 1.94), the mean plasma level of pro-BDNF was 19.63 ng/mL (min = 0.00, max = 162.16, SD = 36.63), and the mean pro-BDNF_0/BDNF ratio was 15.37 (min = 0.00, max = 161.55, SD = 32.13). The mean LEAQ score collected at the 8-month follow-up was 26.93 (min. 7, max. 35, SD 6.55), the mean plasma level of MMP-9 measured at the 8-month follow-up was 144.16 ng/mL (min = 121.92, max = 525.77, SD = 31.14), the mean plasma level of BDNF was 2.02 ng/mL (min = 0.32, max = 13.27, SD = 2.07), the mean plasma level of pro-BDNF was 21.72 ng/mL (min = 1.72, max = 99.87, SD = 21.72), and the mean pro-BDNF/BDNF ratio was 17.52 (min = 0.169, max = 114.1, SD = 26.95). The mean LEAQ score collected at the 18-month follow-up was 32.49 (min. 22, max. 35, SD 3.25), the mean plasma level of MMP-9 measured at the 18-month follow-up was 121.36 ng/mL (min = 0.65, max = 512.33, SD = 98.69), the mean plasma level of BDNF was 2.78 ng/mL (min = 0.09, max = 18.21, SD = 3.59), the mean plasma level of pro-BDNF was 24.57 ng/mL (min = 40.18, max = 220.13, SD = 40.18), and the mean pro-BDNF/BDNF ratio was 24.06 (min = 0.17, max = 220.20, SD = 40.48).

### 2.2. LEAQ Scores; MMP-9, BDNF, and Pro-BDNF Plasma Levels; and Pro-BDNF/BDNF Ratio: Effect of Age at Implantation

Figure 1A–E shows the changes in the mean values of the LEAQ score, the plasma levels of tested proteins, and the pro-BDNF/BDNF ratio within each subgroup and compares the differences between the changes in the mean values of the LEAQ scores, the plasma levels of tested proteins, and the pro-BDNF/BDNF ratio between all follow-up intervals across the subgroups.

The mean LEAQ scores increased significantly across all follow-up intervals in both subgroups. They were significantly different between the subgroups at CI activation—that is, children implanted early had lower LEAQ scores than patients implanted after 1 year old. At the later intervals, the difference lost significance (Table 1). Over the 18 months of follow-up, the absolute changes in the mean LEAQ scores differed significantly between the subgroups so that the younger subgroup showed a significantly greater increase in LEAQ scores (Figure 1A). The MMP-9 plasma concentrations showed a significant decrease between 0 months and the 18-month follow-up and between 0 months and the 8-month follow-up in both subgroups; however, the decrease was significant between the 8- and 18-month follow-ups in the older subgroup and non-significant in the younger subgroup (Figure 1B).

The differences between the MMP-9 levels at all three intervals, as well as absolute changes in the mean values between intervals, were not significant (Table 1 and Figure 1B). The mean values of the BDNF plasma concentrations did not show any significant differences between the follow-up intervals within each of the two subgroups (Figure 1C). They differed significantly between the two subgroups at CI activation and at the 18-month follow-up so that children implanted early had higher BDNF levels than patients implanted after 1 year. At the 8-month follow-up, we did not see any significant difference between the two subgroups (Table 1). However, there was a significant difference in the absolute changes in the mean values of BDNF plasma levels between subgroups between CI activation and the 8-month follow-up (Figure 1C). The changes in the mean pro-BDNF plasma concentrations did not show significant alterations over the follow-up in either of the two subgroups. There were no significant differences between the follow-up intervals or between the mean values at the follow-up intervals (Table 1 and Figure 1D). The pro-BDNF/BDNF ratios showed significant increases between 0 months and the 8-month follow-up and between 0 months and the 18-month follow-ups in the younger implanted group, but no significant changes were observed in the older subgroup. The differences between pro-BDNF/BDNF ratios at all three intervals, as well as the absolute changes in the mean values between the intervals, were not significant (Table 1 and Figure 1E).

### 2.3. MMP-9, BDNF, and Pro-BDNF Levels and Pro-BDNF/BDNF Ratio in the Control Group

In the whole control group, the mean plasma level of MMP-9 was 289.7 ng/mL (min = 36.7, max = 659.0, SD = 179.7), the mean plasma level of BDNF was 3.25 ng/mL (min = 0.3, max = 17.1, SD = 3.2), the mean plasma level of pro-BDNF was 14.1 ng/mL (min = 1.6, max = 110.9, SD = 24.1), and the mean pro-BDNF_0/BDNF ratio was 7.8 (min = 0.1, max = 46.0, SD = 11.4). In the younger control subgroup, the mean plasma level of MMP-9 was 158.89 ng/mL (min = 36.76, max = 340.34, SD = 123.21), the mean plasma level of BDNF was 1.82 ng/mL (min = 0.39, max = 2.78, SD = 1.01), the mean plasma level of pro-BDNF was 9.84 ng/mL (min = 1.65, max = 23.96, SD = 10.44), and the mean pro-BDNF/BDNF ratio was 12.16 (min = 0.62, max = 45.98, SD = 19.35). In the older control subgroup, the mean plasma level of MMP-9 was 304.56 ng/mL (min = 57.63, max = 659.01, SD = 180.08), the mean plasma level of BDNF was 3.41 ng/mL (min = 0.30, max = 17.10, SD = 3.33), the mean plasma level of pro-BDNF was 14.60 ng/mL (min = 1.78, max = 110.93, SD = 25.31), and the mean pro-BDNF/BDNF ratio was 7.32 (min = 0.10, max = 43.51, SD = 10.45).

### 2.4. Comparisons between the Study Group and the Control Group

Due to the small number (N = 5) of the younger control group, we did not perform paired comparisons with it. Instead, we compared the protein plasma levels between the older implanted subgroup and the age-matched controls.

Figure 2 shows that for MMP-9, BDNF, and the pro-BDNF/BDNF ratio, the differences were significant between the older study subgroup and the age-matched control subgroup. The MMP-9 plasma levels were 100.8 ng/mL vs. 304.6 ng/mL (*p* < 0.001), the BDNF plasma levels were 1.8 ng/mL vs. 3.4 ng/mL (*p* < 0.001), the pro-BDNF plasma levels were 18.1 ng/mL vs. 14.6 ng/mL (*p* = 0.07), and the pro-BDNF/BDNF ratios were 27.6 vs. 7.3 (*p* < 0.01).

### 2.5. Correlations between Age and Protein Levels in Both the Study and Control Groups

In order to check whether there was a relationship between the protein levels and age in both the study group and the control group, we tested for correlations between age and the protein levels but did not find any significant results.

## 3. Discussion

In this study, we tried to find differences in the dynamics of auditory development after cochlear implantation in two distinct age groups. In these groups, we explored the trajectories of the MMP-9, BDNF, and pro-BDNF levels, as well as the pro-BDNF/BDNF ratio, measured prospectively at three sequential time points and with simultaneous measurements of auditory development.

The main results included the following findings: (i) There was a significant difference between the study subgroups in terms of the BDNF plasma level at cochlear implantation and at the 18-month follow-up, which was paralleled at cochlear implantation by a significant difference in the LEAQ scores. (ii) For the study subgroups, we observed a significant difference in the changes in BDNF plasma levels from cochlear implantation until the 8-month follow-up, while in terms of LEAQ scores, we also saw a significant difference between the study subgroups in terms of increases from cochlear implantation up to the 18-month follow-up. (iii) There were significant decreases in the plasma levels of MMP-9 from cochlear implantation up to the 18-month follow-up interval and from cochlear implantation to the 8-month follow-up in both subgroups, whereas a significant decrease between the 8- and 18-month follow-up intervals was seen only in the older subgroup. (iv) The mean values of MMP-9, BDNF, and pro-BDNF and the pro-BDNF/BDNF ratio in the older control group were significantly different from the same protein parameters measured in the age-matched older CI subgroup.

### 3.1. Auditory Development and Plasma Protein Levels in Subgroups Implanted before and after 1 Year of Life

Children implanted at less than 1 year old had significantly higher levels of BDNF before their CI was switched on, and this difference was accompanied by a difference in the mean LEAQ score. The evolution of the BDNF levels at further follow-ups was not accompanied by any regular change in the LEAQ score in any of the implanted subgroups (Figure 1A–E). However, it remains unclear whether the initial difference in the BDNF protein plasma concentration between the subgroups had any effect on the early acquisition of auditory skills. Favoring a causal relationship, there are findings that show there is a significantly lower plasma concentration of BDNF protein in children who have been sound-deprived for at least 12 months, which is in line with data from deafened animals [36]. In our study, we found that the LEAQ scores at the 8-month follow-up did not differ significantly between the subgroups, although there was a significant difference in the change (in the opposite direction) in the mean BDNF plasma level between CI activation and the 8-month follow-up. It therefore remains unclear whether the significantly greater increase in the LEAQ score between 0 month and the 18-month follow-up in the younger implanted subgroup can be connected with an initial decrease in the BDNF plasma level from 0 month to the 8-month follow-up.

Moreover, the difference in the pro-BDNF/BDNF ratio between the two subgroups did not correspond in any simple way to the auditory development measures. However, when looking at the changes over time in the MMP-9 plasma level and the pro-BDNF/BDNF ratio in both subgroups, one can see that the plasma level of MMP-9 decreased significantly from cochlear implantation to the 8-month follow-up and from the 8-month follow-up to the 18-month follow-up. The difference was in the 8- to 18-month follow-up interval, where there was a (non-significant) decrease in the younger subgroup. This pattern of relations of the MMP-9 plasma level with auditory development can be interpreted in analogy with the already reported significant correlation between the MMP-9 plasma level measured at cochlear implantation and the LEAQ score at the 18-month follow-up (in an older implanted group), whereas there was a lack of a significant correlation in a younger implanted group [12]. Our finding here builds on our earlier retrospective study of a group of congenitally deaf children implanted up to 2 years of life, where we reported that carriers of the less transcriptionally active C/C genotype rs3918242 of the *MMP9* gene had higher auditory development scores after 24 months of rehabilitation [35].

Our results add to accumulating evidence that the age at which sensory stimulation is delivered to the auditory cortex is associated with different molecular machinery: there is a real difference in neuronal plasticity after congenital deafness treatment with cochlear implantation compared to normal-hearing subjects [35]. As neurodevelopmental processes take some time, it seems justified to look at LEAQ scores in the light of molecular changes that occurred prior to the measured clinical outcomes. However, the current clinical dataset and the confirmed predictive value of MMP-9 as a peripheral biomarker do not allow any firm conclusions to be drawn about specific molecular processes in which MMP-9 may be involved, particularly about the auditory development of children implanted very early vs. children implanted after their first birthdays.

Similar data on auditory development reported by other authors have shown, for the period 6–8 months after CI activation, greater differences between the trajectories of language acquisition of children implanted before and after 1 year of age; however, in these reported cases, the groups were appreciably smaller and of unknown etiology [8,10]. Moreover, in these same earlier studies, the language development trajectories of children implanted before and after 1 year of age had already plateaued by 18 months of CI use, so the authors did not see any significant difference in the LEAQ scores. This perspective is in line with our data.

### 3.2. BDNF Plasma Level Variation

Both our subgroups started with significantly different plasma levels of BDNF, and the levels changed in different ways over time. Both these observations can be interpreted in terms of the natural dynamics of BDNF plasma levels in different age groups as well as reflecting changes in BDNF plasma concentrations due to auditory deprivation and subsequent CI stimulation [36,37]. A literature search failed to return data on BDNF plasma level variation over time in healthy human neonates or toddlers, so here we have to rely on data from animals and human adolescents and adults. However, the results of a large study of healthy human plasma proteomics reported by Bjelosevic et al. [38] indicated that the levels of most of proteins detected in neonate plasma are very distinct from those detected in other age groups, such as children aged 1–5 years old, adolescents, and adults. Unfortunately, our data from normal-hearing controls do not allow us to make conclusions about the natural dynamics of the measured proteins, so it is unclear if the lower plasma level of BDNF identified in the older subgroup at cochlear implantation is in any way connected with the longer absence of auditory stimulation. However, Tan et al. [36] reported a relative decrease in BDNF concentration in the brain tissue of deafened animals compared to those with normal hearing. This finding, together with the results of other animal studies reporting that BDNF plasma levels reflect its concentration in brain tissue [39], can be interpreted as supporting significantly lower mean BDNF plasma levels in our older subgroup since longer auditory deprivation might be expected to lead to a decrease in the plasma BDNF level [37]. The subsequent trend of an increase in neurotrophin concentration in the older subgroup also corresponds with the findings of Tan et al., which showed an increase in BDNF levels in brain tissue 7 weeks after cochlear implantation. Data from adolescents and adults on plasma BDNF levels indicate that neurotrophin plasma concentrations tend to decrease with age [40]. On the other hand, Alemi et al. [34] reported that the BDNF plasma levels in a group of 15 deaf children were much higher than those seen in our data: at cochlear implantation at 3 years old (mean age 36.8 months), the children had a mean BDNF protein plasma level of 58.1 ng/mL, and this increased to 79.5 ng/mL 7 months after the intervention. In their study, the authors did not report deafness etiology or duration, and the authors did not clearly indicate whether their ELISA kit differentiated pro-BDNF from BDNF, so for these reasons their results cannot be directly compared to ours.

### 3.3. Blood–Brain Barrier for BDNF and MMP-9 after Auditory Deprivation and after CI

Much is known about the regulation of MMP-9 and BDNF in neuron bodies and synapses [13,18,22], their transportation across the blood–brain barrier (BBB) [41,42,43,44], and the clinical relevance of both molecules to auditory development in implanted children [12]. Nevertheless, we still do not know the exact mechanism by which the plasma levels of the proteins control synaptic dynamics in the brain neuropil. It has been reported that both molecules can also be found in vascular endothelial cells [18,41,42], leading one to think that at least part of the circulating MMP-9, BDNF, and pro-BDNF comes from brain tissue via the permeability of the BBB. The activity of MMP-9 and BDNF takes place mainly within excitatory synapses, which settle on the dendritic spines of neurons in the infragranular layer [13,37,45]. Data from deafened cats showed that sensory deprivation led to a series of severe molecular consequences in and between neurons, such as insufficiency in neurotransmitter production or sequestration (such as, for example, a decrease in BDNF expression) [36,37,45,46,47]. This significantly reduced synaptic density and excitability in the infragranular layer of the auditory cortex and dissipation of the layer itself [48,49]. A reduction in infragranular layer activity has a considerable impact on cortical processing during later development, including on feedback projections of the primary and secondary auditory cortices [49]. Applied to our study, in the group of older implanted children who were deprived of sound up to the second year of life, the low level of BDNF protein in plasma at cochlear implantation could then possibly be interpreted to be due to an insufficiency of the infragranular layer as a result of poor neural stimulation. The question remains as to whether the low level of BDNF is a consequence of the degeneration of the pathway, its abnormal maturation, or both. As for why the difference in MMP-9 protein plasma levels between both groups was non-significant, only tentative answers are possible. The observed variations might be interpreted in terms of the permeability of the BBB to MMP-9, which decreases with the age of the patients [43]. In addition, variations over the follow-ups may reflect the activity and regulation of MMP-9, which may, in some way, depend on the biological age of the implanted child or perhaps on the electrical stimulation supplied by the CI [13,14,19].

### 3.4. Comparison with Control Group

Figure 2 compares the mean values of MMP-9 and BDNF and indicates significantly higher concentrations of these proteins in the healthy controls than in CI children in the subgroup implanted after their first birthdays; similarly, the pro-BDNF/BDNF ratio was significantly lower in the controls. Due to the difficulty of enrolling young healthy children, the control group for the younger implanted subgroup was under-represented, so all conclusions based upon these results are necessarily questionable. A lack of prospective longitudinal observations in the control group and no correlation of proteins levels with age also make firm conclusions difficult.

### 3.5. Perspective

Our findings that MMP-9 and BDNF plasma levels play an important role in governing auditory development following deafness treatment are necessarily tentative and require verification. Longer observation periods would be beneficial. In particular, if the plasma levels of BDNF correlate with LEAQ scores over longer time frames, this would be an encouraging result. In such a longer-term study, a more reliable test of speech understanding could be used to replace the less reliable tool (LEAQ) that we had to use to assess auditory development in infants and toddlers.

### 3.6. Limitations

Due to the relatively short observation time and sparse follow-up intervals, our findings need to be interpreted with considerable caution. In addition, assessing auditory development using a parental questionnaire such as LEAQ carries a considerable degree of subjectivity. Another subjective factor that is difficult to control is the effect of the child’s environment, including family support, parental involvement, and motivation to support the child’s efforts [7]. The greatest caveat is the absence of prospective observations of the control group, meaning that the interpretation of the results rests on considerable individual variability. On the other hand, obtaining such data would be extremely difficult; nevertheless, it would be extremely interesting to see the natural variation in the tested proteins over time in heathy controls and compare it with that in an implanted cohort. A larger study group would add statistical weight to the results and increase the strength of the conclusions.

## 4. Materials and Methods

### 4.1. Study Design, Participants, and Ethical Approval

This prospective study was carried out at the Institute of Physiology and Pathology of Hearing in Warsaw, Poland, between December 2016 and December 2019. A group of infants and toddlers with congenital deafness was enrolled to the study group and underwent routine cochlear implantation. The procedure was performed by the same surgeon with the same type of device. The inclusion criteria were congenital bilateral profound sensorineural hearing loss, confirmed bilaterally with auditory brainstem responses (ABRs) over 80 dB, and the activation of a speech processor before the age of 2. The exclusion criteria were chronic concomitant disease, the presence of any acute inflammation confirmed by C-reactive protein (CRP) measurements, a history of severe prematurity, asphyxia, or Toxoplasmosis, Other, Rubella, Cytomegalovirus, Herpes Simplex (TORCH) infection during pregnancy. After device activation, parents or caregivers followed instructions on auditory–verbal therapy. There were three prospective observation points: at cochlear implantation and at 8 and 18 months after CI activation. At each interval, measurements were made of the MMP-9, BDNF, pro-BDNF, and CRP plasma levels (MMP-9_0, MMP-9_8, and MMP-9_18; BDNF_0, BDNF_8, and BDNF_18; and pro-BDNF_0, pro-BDNF_8, and pro-BDNF_18) as well as auditory development measurements performed with the LittlEARS Auditory Questionnaire (LEAQ_0, LEAQ_8, and LEAQ_18). Initially, 70 children were enrolled, but this was reduced to 59 because of parents withdrawing from the study (2 cases), an autism spectrum disorder diagnosis over the follow-up period (1 case), elevated CRP levels (6 cases), and subjects being lost from follow-up (2 cases). The final group of 59 children was divided into two subgroups according to their age at CI activation: before 1 year old (N = 29) and after 1 year old (N = 30). In the first subgroup, there were 13 girls (45%) and 16 boys (55%), and the mean age at CI activation was 9.9 months (min = 6.9, max = 12.1, SD = 1.5). In the latter subgroup, there were 14 girls (47%) and 16 boys (53%), and the mean age at CI activation was 16.6 months (min = 12.3, max = 23.7, SD = 3.2). All enrolled children were of Caucasian origin, were implanted with the Med-El Synchrony CI, and became regular CI users. We successfully collected demographic data, blood plasma samples, and LEAQ scores from all participants at all three time points.

The control group included 49 children who underwent surgery at the Institute of Physiology and Pathology of Hearing for adenoid or palatine tonsil hypertrophy, with normal hearing confirmed by evoked otoacoustic emissions and impedance tympanometry. The exclusion criteria were the same as for the study group. The ages in the control group were matched to the ages the children enrolled in the study group were at the 18-month follow-up. The control group was subjected to one measurement of MMP-9, BDNF, pro-BDNF, and CRP plasma levels. Initially, there were 59 in the control group, but this was reduced to 49 due to elevated CRP levels. There were 25 girls (51%) and 24 boys (49%). The mean age at the measurement of plasma protein levels in the final control group was 36.3 months (min = 26.6, max = 41.9, SD = 3.7). We divided the group into two subgroups. The ages of the younger control group (N = 5) ranged from 26.6 to 29.6 months and were matched to the ages of the “CI activation before 1 year old” group at their 18-month follow-up (from 24.5 to 31.2 months); similarly, the ages of the older control group (N = 44) ranged from 31.2 to 41.9 months and were matched to the ages of the “CI activation after 1 year old” group at their 18-month follow-up (from 30.4 to 41.9 months).

### 4.2. Auditory Development Assessment

We used the LittlEARS Auditory Questionnaire (LEAQ) to assess the auditory development of the implanted children in our cohort [50]. LEAQ consists of 35 questions with “yes” or “no” answers and has been validated in many other languages. The total score is the number of “yes” answers [51,52,53,54].

### 4.3. Plasma Sample Collection

Blood samples (4 mL) were collected from the ulnar vein with heparin as an anticoagulant. After sampling, tubes were centrifuged for 15 min at 1400× *g*. Plasma was obtained, aliquoted, and stored at −80 °C for further analysis. The total protein content was estimated using a BCA protein assay kit (Thermo Fisher Scientific, Carlsbad, CA, USA) following the manufacturer’s protocol.

### 4.4. Plasma MMP-9, BDNF, and Pro-BDNF Levels

The plasma concentrations of MMP-9, BDNF, and pro-BDNF were measured using specific enzyme-linked immunosorbent assay (ELISA) kits (MMP-9 and BDNF—R&D Systems Inc., Minneapolis, MN, USA; pro-BDNF—Aviscera Bioscience, Santa Clara, CA, USA). ELISAs were performed according to the manufacturer’s protocol. A total of 30 μg/μL of protein from each plasma sample was diluted 70-fold (MMP-9) or 20-fold (BDNF and pro-BDNF) with calibration diluent from the assays and analyzed in duplicate. The optical density of wells was determined at 450 nm using an automated microplate reader (Sunrise Microplate Absorbance Reader, Tecan, Männedorf, Switzerland).

### 4.5. Statistical Analyses

#### 4.5.1. Paired Comparisons

For all tested follow-up intervals (i.e., at CI activation and at 8 and 18 months after CI activation), comparisons were made for the mean LEAQ scores; the MMP-9, BDNF, pro-BDNF levels; and the pro-BDNF/BDNF ratio between patients belonging to the two study subgroups and between the respective patient and control groups using a Welch two-sample *t*-test or a Wilcoxon rank-sum test. All calculations were performed with the R language (version 3.6.3). The results were considered statistically significant at a *p*-value < 0.05.

#### 4.5.2. Tests for Significance of Differences between Changes in LEAQ Scores; Plasma Levels of MMP-9, BDNF, and Pro-BDNF; and Pro-BDNF/BDNF Ratio

Differences between the values of variables at different intervals were calculated by subtracting the value in the previous interval from the value in the next. For example, LEAQ_change0–8_ = LEAQ_8_ − LEAQ_0_ so that positive values indicate an increase in the value of the variable between the two intervals and negative values indicate a decrease.

For all tested follow-up intervals, the comparisons of mean LEAQ scores; BDNF, pro-BDNF, and MMP-9-levels; and pro-BDNF/BDNF ratios were made between patients belonging to the two study subgroups and the control group using a Welch two-sample *t*-test (if test assumptions were met) or a Wilcoxon rank-sum test (if not). All calculations were performed with R (version 3.6.3). The results were considered statistically significant at a *p*-value ≤ 0.05.

#### 4.5.3. Assessment of the Significance of Changes in Values in Subsequent Measurements for LEAQ, MMP-9, BDNF, Pro-BDNF, and Pro-BDNF/BDNF Ratio in Subgroups

For each of the subgroups, the total significance of the changes in values over time was assessed as well as the change between the two respective measurements (0–8, 8–18, and 0–18). An analysis of variance (ANOVA) was used to assess the cumulative significance of changes in successive measurements if the assumptions were met, and Friedman’s ANOVA was used when at least one assumption was not met. Post hoc tests with the Bonferroni correction were used to compare the mean differences in the two respective measurements: *t*-tests for paired samples if the assumptions were met and the Wilcoxon signed-rank test otherwise. The results were considered statistically significant at a *p*-value ≤ 0.05.

#### 4.5.4. Correlation Analysis

The age at the time of testing; the MMP-9, BDNF, and pro-BDNF levels; and the pro-BDNF/BDNF ratio were measured at the 18-month follow-up interval in the study group, and the age at the time of testing and the same protein levels were measured in the control group and tested for their levels of correlation using Pearson or Spearman tests. Prior to correlation testing, a Shapiro–Wilk test of normality was conducted in order to check assumptions. All variables for which the correlations were tested were normalized using the min–max scaling method. A correlation was considered statistically significant at a *p*-value ≤ 0.05. All computations were made using R version 3.6.3 (2020).

## Figures and Tables

**Figure 1 ijms-24-03714-f001:**
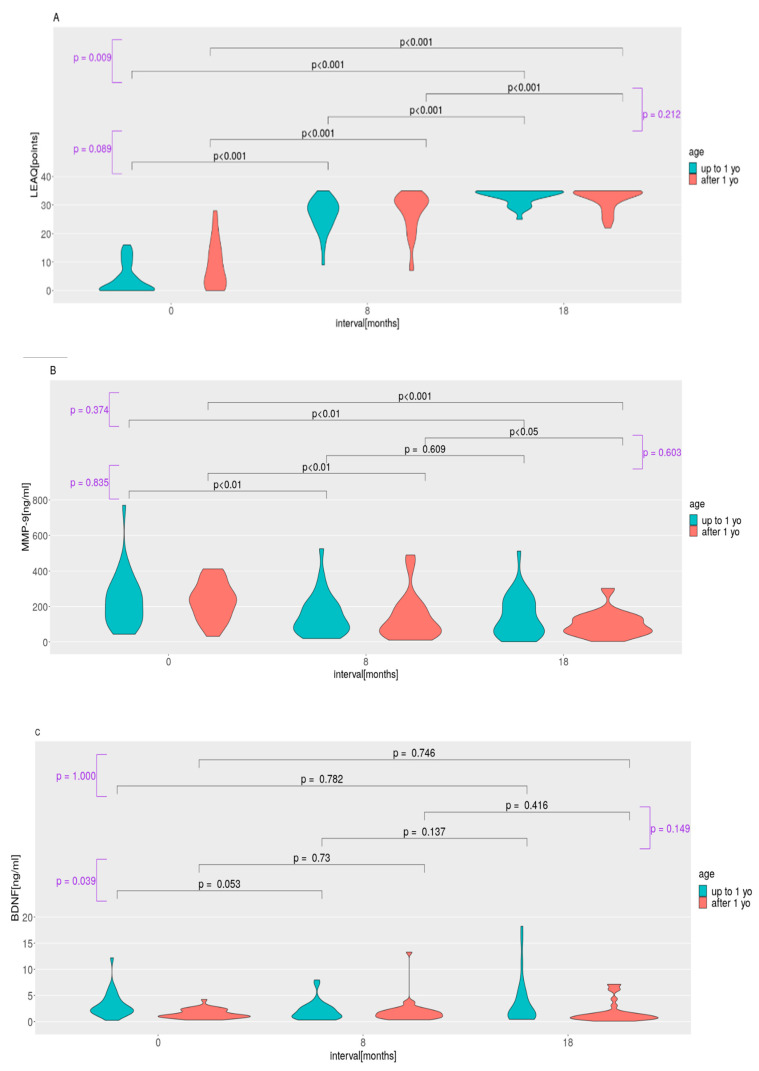
LEAQ score (**A**) and plasma levels of MMP-9 (**B**), BDNF (**C**), and pro-BDNF (**D**) as well as the pro-BDNF/BDNF ratio (**E**) at all three follow-up intervals in the two study groups: one in which CI activation occurred before 1 year (N = 29, cyan) and the other where CI activation took place after 1 year (N = 30, red). There were differences in the changes in the means of LEAQ, MMP-9, BDNF, and pro-BDNF, as well as the pro-BDNF/BDNF ratio, between the subgroups over the follow-up intervals (*p*-values, violet) as well as changes in the means of LEAQ, MMP-9, BDNF, and pro-BDNF as well as the pro-BDNF/BDNF ratio over the follow-up within both subgroups (*p*-values, black). Statistical differences are shown with *p*-values at the top.

**Figure 2 ijms-24-03714-f002:**
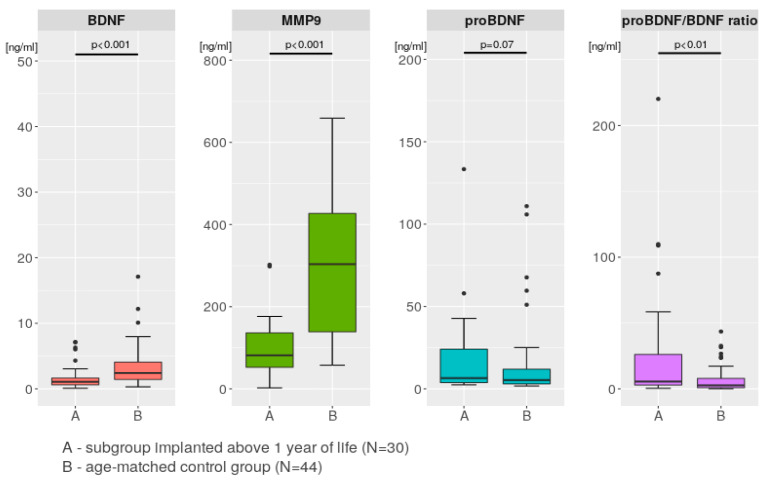
Paired comparisons of the mean plasma levels of MMP-9_18, BDNF_18, and pro-BDNF-18, as well as the pro-BDNF_18/BDNF_18 ratio, measured in the study subgroup implanted after 1 year of life (A, N = 30) and in the age-matched control group (B, N = 44).

**Table 1 ijms-24-03714-t001:** LEAQ scores and plasma levels of MMP-9, BDNF, and pro-BDNF, as well as the pro-BDNF/BDNF ratio, at all three follow-up intervals in the two study groups: one in which CI activation occurred before 1 year (N = 29) and the other where CI activation took place after 1 year (N = 30). Paired comparisons are of the mean values between the groups.

CI Activation	LEAQ_0				LEAQ_8				LEAQ_18			
	Mean	Min	Max	SD	Mean	Min	Max	SD	Mean	Min	Max	SD
<1 y. o.	3.93	0	16	5.13	26.55	9	35	5.87	32.90	25	35	2.64
>1 y. o.	8.73	0	28	8.10	27.30	7	35	7.24	32.10	22	35	3.75
*p*-value	<0.01				0.277				0.619			
	MMP-9_0 (ng/mL)				MMP-9_8 (ng/mL)				MMP-9_18 (ng/mL)			
	mean	min	max	SD	mean	min	max	SD	mean	min	max	SD
<1 y. o.	235.03	43.3	769.6	151.7	150.7	19.0	525.7	116.69	142.6	0.67	512.3	118.2
>1 y. o.	231.1	31.1	412.0	105.1	137.7	10.3	490.4	128.4	100.78	2.39	302.09	71.38
*p*-value	0.712				0.435				0.292			
	BDNF_0 (ng/mL)				BDNF_8 (ng/mL)				BDNF_18 (ng/mL)			
	mean	min	max	SD	mean	min	max	SD	mean	min	max	SD
<1 y. o.	3.13	0.25	2.18	2.38	2.08	0.32	7.95	1.84	3.76	0.45	18.21	4.50
>1 y. o.	1.51	0.32	4.24	0.86	1.96	0.36	13.27	2.31	1.85	0.09	7.14	2.10
*p*-value	<0.001				0.682				<0.05			
	pro-BDNF_0 (ng/mL)				pro-BDNF_8 (ng/mL)				pro-BDNF_18 (ng/mL)			
	mean	min	max	SD	mean	min	max	SD	mean	min	max	SD
<1 y. o.	16.98	1.41	158.6	32.66	20.74	2.52	99.78	24.47	31.27	1.82	220.13	50.57
>1 y. o.	22.18	0.00	62.16	40.50	13.59	1.72	75.29	18.42	18.09	2.47	133.38	25.92
*p*-value	1.000				0.126				0.481			
	pro-BDNF_0/BDNF_0 ratio				pro-BDNF_8/BDNF_8 ratio				pro-BDNF_18/BDNF_18 ratio			
	mean	min	max	SD	mean	min	max	SD	mean	min	max	SD
<1 y. o.	10.54	0.20	102.8	22.48	21.84	0.35	114.0	30.76	20.43	0.17	127.19	31.29
>1 y. o.	20.03	0.00	161.5	39.12	13.35	0.19	108.5	22.40	27.56	0.39	220.2	48.02
*p*-value	0.097				0.386				0.484			

min—minimum value; max—maximum value; SD—standard deviation.

## Data Availability

The datasets analyzed during this study are accessible on reasonable request.

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
