# Peer review of "Longitudinal Changes in BDNF and MMP-9 Protein Plasma Levels in Children after Cochlear Implantation"

_ijms, 2023, doi:10.3390/ijms24043714_

Round 1

Reviewer 1 Report

The paper " Longitudinal changes of BDNF and MMP-9 protein plasma levels in children after cochlear implantation" by Matusiak et al. compared several protein concentrations in the bloodstream in children whose CI was activated before (young) or after one year old (old). They have two aims: (i) To test whether plasma levels of MMP-9, BDNF, and pro-BDNF/BDNF ratio are significantly different between young and old children. (ii) To elucidate whether the same parameters measured at CI use are similar to those in an age-matched control group of normal-hearing children. The study provides important information to the literature. The experimental design is straightforward and well-explained. However, there are many confusing and contradicting statements in the text.
1. The abstract lines 22-24 stated, "Between the subgroups there were significant differences between changes of BDNF levels from 0 to 8 months follow-up, and LEAQ scores from 0 to 18 months follow-up."  Did the authors get this conclusion from Figure 1C and Figure 1A? Fig 1 is very confusing. For examples, for example, Fig1A has nine p-values. All of them are <0.05 except two, p=0.0089 and p=0.212. I have no idea where these numbers come from ( for which subgroup pairs). If my guess is right, I got the totally opposite conclusion. For the same reason, I did not understand the Figure 1C. In lines 127-128, it stated, "However, there was a significant difference between the absolute change of the mean values of BDNF plasma levels between CI activation and 8 months follow-up (Fig. 1C)." I assume that authors are describing the young group here. But the p-value between CI activation and 8-month follow-up subgroup is 0.053. Did I miss something?

2. The abstract lines 26-26 stated, "For all measured protein concentrations, significant differences were identified between the study group and the control group." This conclusion is inaccurate because only the old study group was compared with the control group.

3. Reference 12 and 33 are almost identical. Any reason for separating them into 2 references?

4. What is the reason for measuring pro-BDNF/BDNF ratio?

5. Line 89. In this result section, there is no description of LEAQ scores at all. Also, what is the reason to list the means for "whole study group". This group data is not used to compare with the normal hearing control. etc.

6. Lines 95-97 stated, "The mean plasma level measured at the 8 months follow-up was 144.16 ng/ml (min = 121.92, max = 525.77,  SD = 31.14), " Should I assume it is for MMP-9?

7.  Children in the young group were at least six months younger than the old group. The authors did not discuss the potential age effect on LEAQ. Based on the limited data from the normal hearing control group, the concentrations of MMP-9, BDNF, etc., also differ between young and old children. Thus, age could have an important impact on these proteins' concentrations in the bloodstream. It is hard to get a clear conclusion without the normal young control data.

8. Please explain the p values that were not on the top of the lines in Fig. 1.

9. What is "CRP" stand for?

10. Does ELISA affected by hemolysis?

11. For ELISA, was equal amount of protein used for ELISA, or an equal volume of plasma was used for each sample?

Author Response

Please, find attached our response to the Reviewer.

Reviewer 2 Report

The study reported changes in biomarkers after cochlear implant in children, including MMP-9, BDNF, pro-BDNF levels and pro-BDNF/BDNF ratio. The results can be useful in quantifying different outcomes after cochlear implant. The study is clearly presented.

The manuscript could be improved by organizing the main sections in Introduction, Materials and methods, Results and Discussions - it would be more easy to follow.

Author Response

Please find attached our response to the Reviewer.

Round 2

Reviewer 1 Report

Response to the cover letter ( in red):

1. The abstract lines 22-24 stated, "Between the subgroups there were significant differences between changes of BDNF levels from 0 to 8 months follow-up, and LEAQ scores from 0 to 18 months follow-up."  Did the authors get this conclusion from Figure 1C and Figure 1A? Fig 1 is very confusing. For examples, for example, Fig1A has nine p-values. All of them are <0.05 except two, p=0.0089 and p=0.212. I have no idea where these numbers come from ( for which subgroup pairs). If my guess is right, I got the totally opposite conclusion.

We are grateful to the reviewer for this comment. Indeed, Figure 1 appears to be somewhat difficult for interpretation, therefore we have improved it by changing the color of p-values of  differences between changes of mean values between intervals between subgroups. They are marked as violet. The p-values of changes of mean values between intervals within each subgroup remain black. It has been also explained in the Figure 1 caption.

No, it is still confusing. I cannot understand where these violet p values came from. They are not the p-values between the 2 study groups (young and old) in three follow-up intervals, which are already shown in Table I.

Black p-value is easy to follow because it used lines to indicate the compared pairs. If you could use lines to indicating compared pair for violet p-values, that will be better. Or make a new graph to show the changed values of between intervals between subgroups. It is very hard for readers to following your data.

Figure 1 caption is even worse. Where can I find blue p-value?

2. For the same reason, I did not understand the Figure 1C. In lines 127-128, it stated, "However, there was a significant difference between the absolute change of the mean values of BDNF plasma levels between CI activation and 8 months follow-up (Fig. 1C)."   I assume that authors are describing the young group here. But the p-value between CI activation and 8-month follow-up subgroup is 0.053. Did I miss something?

We thank the Reviewer for this comment. Indeed, the current sentence should be elaborated on. We have corrected it: „However, there was a significant difference between the absolute change of the mean values of BDNF plasma levels between subgroups between CI activation and 8 months follow-up (Fig.1 C).„

See the reason above.

4. What is the reason for measuring pro-BDNF/BDNF ratio?

We are grateful to the Reviewer for asking this question. We have added the explanation of mutual relations between BDNF, pro-BDNF and MMP-9 in the Introduction:                                            “BDNF is cleaved to produce mature form of BDNF and MP-9 has been implicated in this conversion [1].  Pro-BDNF/BDNF ratio has been hence analyzed in clinical samples of e.g., schizophrenia [2]. “

1.       Khan M. Z., Zheng Y. B., Yuan K., Han Y., Lu L. Extracellular zinc regulates contextual fear memory formation in male rats through MMP-BDNF-TrkB pathway in dorsal hippocampus and basolateral amygdala. Behav Brain Res. 15;439:114230 (2023)

2.       Yamamori H., Hashimoto R., Ishima T., Kishi F., Yasuda Y., Ohi K., Hujimoto M., Umeda-Yano S., Ito A.,Hasshimoto K., Takeda M. Plasma levels of mature brain-derived neurotrophic factor (BDNF) and matrix metalloproteinase-9 (MMP-9) in treatment –resostent schizophrenia treated with clozapine. Neurosci. Lett., 27; 556: 37-41 (2013),     

OK. But you need to pay attention to punctuation markers in the paper.
5. Line 89. In this result section, there is no description of LEAQ scores at all. Also, what is the reason to list the means for "whole study group". This group data is not used to compare with the normal hearing control. etc.

Thank You for this question. We have listed the means for the study group (Results 2.1) because in order to check age effect on MMP-9, BDNF, pro-BDNF levels and pro-BDNF/BDNF ratio in normal hearing controls and in implanted children we have tested for correlations between age and the proteins (Results 2.5) in the whole study group and in the whole control group.

Original data are always valuable. Since you only compared the old group with normal control , why do you listed them in a table, which include data of  the whole group, old group, and young group?

7.  Children in the young group were at least six months younger than the old group. The authors did not discuss the potential age effect on LEAQ.

We are grateful to the Reviewer for this question. Validation study of the LEAQ in children with cochlear implants performed on the larger group of 122 CI children, who met the same inclusion criteria as in our study, reported that there was no correlation of the total score at CI activation with chronological age. Although the authors reported 4 point difference in total score between early implanted children (10 months) and late implanted (16 months), they did not found it to be significant [3]. In our study we observed differences in the LEAQ score between older and younger group of patients only at the time of cochlear implantation. In next months there were no significant differences between these subgroups. Therefore we did not tested for correlation between age and LEAQ 0.  

3.       Obrycka A., Lorens A., Padilla J.L., Piotrowska A., SkarżyÅ„ski H. (2017) Validation of the LittlEARS Auditory Questionnaire in cochlear implanted infants and toddlers. International Journal of Pediatric Otorhinolaryngology, 93:107-116. doi: 10.1016/j.ijporl.2016.12.024

4.       OK

Based on the limited data from the normal hearing control group, the concentrations of MMP-9, BDNF, etc., also differ between young and old children. Thus, age could have an important impact on these proteins' concentrations in the bloodstream. It is hard to get a clear conclusion without the normal young control data.

We are grateful to the Reviewer for this comment. We would like to point out, again, that have tested the correlations between age and the proteins plasma levels in both the whole study group and the whole control group and no significant results were found. (Results 2.5)

Yes, it is not surprised that the total protein levels in the plasma do not change much, similar to previous reported (Hickmans et al., 1943). However, individual proteins could change dramatically. See recent paper by Bjelosevic et al., 2017: Quantitative Age-specific Variability of Plasma Proteins in Healthy Neonates, Children and Adult.

https://www.sciencedirect.com/science/article/pii/S1535947620323963

As the authors point out, BNDF is very important for nerve development. Young infants could have more BDNF in their bloodstream than old infants do. In the abstract, authors claimed “We identified statistically higher BDNF levels at the 0 and 18 months follow-ups in the younger subgroup compared 21 to the older one…” Because the control data (normal BDNF levels) for the young group is absent, the data to support this conclusion is weak. Authors can add more data or soften the conclusion.

8. Please explain the p values that were not on the top of the lines in Fig. 1.

Thank You for this question. The p-values that were by the Figure’s 1 side show statistical significance of differences between changes of values between relevant intervals between subgroups. In order to make them more visible now they are colored.

See above.

Author Response

Plaese atahced find the responses to the Reviewer.

Round 3

Reviewer 1 Report

Figures are now understandable.